# Validation of a Novel Suprathreshold Strategy for Screening Visual Function

**DOI:** 10.3390/jcm14051526

**Published:** 2025-02-25

**Authors:** Juan D. Arias, Reena Chopra, Mary K. Durbin, Kristen N. Knight, Derek Y. Ho, Marco A. Miranda, Huiyuan Hou, Christina Forte, James L. Fanelli

**Affiliations:** 1Topcon Healthcare Inc., Oakland, NJ 07436, USA; rchopra@topcon.com (R.C.); kknight@topcon.com (K.N.K.); dho@topcon.com (D.Y.H.); mestevesmiranda@topcon.com (M.A.M.); hhou@topcon.com (H.H.); 2Cape Fear Eye Institute, Wilmington, NC 28412, USA; cfortejones@rocketmail.com

**Keywords:** standard automated perimeter, glaucoma, suprathreshold, visual field

## Abstract

**Background/Objectives**: Glaucoma, a leading cause of blindness worldwide, is often associated with high intraocular pressure (IOP), which eventually leads to loss of retinal ganglion cells and the retinal nerve fiber layer. Visual field (VF) testing is a principal method of diagnosing and monitoring this disease. Suprathreshold VF test programs are quicker than threshold strategies and are often used as a screening tool. This study evaluates the TEMPO/IMOvifa (Topcon Healthcare/CREWT Medical Systems, Tokyo, Japan), a bilateral standard automated perimeter with a suprathreshold screening program by assessing the sensitivity in a glaucoma cohort and the specificity in a healthy cohort. **Methods**: All subjects were tested at a single site and underwent a comprehensive ocular examination to categorize them into either a healthy or glaucoma group. As part of the testing procedure, two TEMPO suprathreshold VFs were conducted in sequence and accompanied by a threshold VF test. **Results**: A total of 193 eyes (randomized study eye) (193 subjects) were evaluated in the final analysis (101 healthy and 92 glaucoma), and average suprathreshold test time (SD) per eye was 39.4 (±4.86) seconds. Specificity was at 91% in the healthy group and sensitivity was at 49% in the glaucoma group. Sensitivity was at 100% when applied to glaucoma cases with an MD of less than −3 dB. **Conclusions**: The TEMPO screening program demonstrated strong specificity in detecting true healthy cases. It also demonstrated a strong sensitivity when screening mild to moderate glaucoma. Early glaucoma and glaucoma suspects would benefit from complementary modalities such as optical coherence tomography and threshold perimetry to effectively diagnose. Utilizing this screening program in optometric and ophthalmic settings could yield benefits for both the practitioner and the patient.

## 1. Introduction

Glaucoma is the second leading cause of blindness worldwide and it is estimated that the epidemiological growth rate of this disease will continue at an expansive rate [1]. The disease is a collection of degenerative ocular neuropathies responsible for the loss of retinal ganglion cells and retinal nerve fiber layers, directly impacting the health of the optic nerve head [2,3]. It is clinically characterized by neural tissue loss of the nerve head, excavation and thinning of the cup, as well as non-specific visual field defects [4].

Standard automated perimeters (SAPs) are a principal diagnostic and progression monitoring tool for assessing visual field (VF) loss, alongside optical coherence tomography (OCT), a structural diagnostic device [5]. VF testing quantitatively measures the sensitivity of a patient’s central and peripheral vision to detect localized functional loss. SAP threshold strategies quantify visual loss by adjusting stimulus intensity based on patient responses [6]. Multiple tests are commonly performed over a period of time to measure progression and stability against the variability of the test [7]. Duration of testing depends on the type of test and severity of disease, but is generally more than a few minutes per eye. Fatigue and learning effects can reduce reliability of the visual field due to the long nature of most VF tests, limiting implementations in population-based screening environments [6]. Strategies such as suprathreshold (screening) tests have been used in parallel with threshold testing due to the quicker nature of the program [7,8,9]. In this paper, ‘screening’ and ‘screening test’ refer to the suprathreshold perimetric strategy of assessing visual function.

There are a variety of screening patterns currently being used which differ in methods and serve different functions. Confrontation screening tests are a quick, deviceless method for finding gross functional defects [10]. Frequency Doubling Technology and Kinetic perimetry are useful for detecting early visual function loss and neurological field defects, respectively [11,12]. In two-zone VF tests, test locations are classified into ‘seen’ and ‘not seen’, based on the response to the stimulus. Unlike threshold testing, these tests provide a binary classification of visual function rather than providing a stimulus intensity. Suprathreshold programs simplify VF testing patterns which reduce test burden and increase patient cooperation. Unlike threshold testing, many suprathreshold strategies present stimuli at a fixed intensity, though some may increase intensity after an initial miss to confirm a defect. The stimuli presented in suprathreshold testing are above the expected threshold, and algorithms vary as to how frequently these stimuli are presented or repeated. An advantage to a quicker VF test is that it can be performed more frequently and be implemented into a healthcare setting more easily; therefore, enabling implementation in ocular-based screening environments, which could address the unmet need of identifying undiagnosed disease. These advantages could facilitate earlier disease detection and treatment in populations that are currently underserved or less frequently tested.

The TEMPO/IMOvifa (CREWT Medical Systems, Tokyo, Japan) visual field device is a bilateral standard automated perimeter capable of performing threshold and suprathreshold VF testing on both eyes independently but simultaneously. The device’s suprathreshold screening program utilizes a two-zone 28-point algorithm based off points most correlated with glaucomatous functional loss (Figure 1) [13,14]. The screening threshold tested is based on each location’s normative values relative to age and the tested stimulus threshold is based on the first percentile of the reference database in relation to age. The suprathreshold strategy has a test time range of 33–121 s per eye when tested in a mixed health population which includes advanced glaucoma [14]. The TEMPO screening test is intended to serve as a triage tool to identify individuals with potential visual function loss for referral rather than acting as a standalone diagnostic tool.

The objective of this study is to evaluate the diagnostic accuracy of the TEMPO screening test by measuring the sensitivity in a sample of eyes with glaucoma and the specificity in a sample of healthy eyes.

## 2. Materials and Methods

All participating subjects signed an informed consent form and fulfilled all inclusion and exclusion criteria. Subjects were seen at a single site in the US by an optometrist and underwent a comprehensive ophthalmic exam. The Advarra Institutional Review Board (6100 Merriweather Dr, Suite 600, Columbia, MD, USA) approved the study protocol, and the methodology adhered to the tenets of the Declaration of Helsinki for research involving human subjects and to the Health Insurance Portability and Accountability Act.

### 2.1. Study Participants

The study was prospective in nature and took place at a single site in Wilmington, North Carolina. The cohort comprised two groups: healthy and glaucoma. Subjects were assigned to their pertaining group based on either a comprehensive examination performed by the optometrist with VF damage or structural loss providing additional diagnostic confirmation or based on findings from their past ocular examinations.

Included subjects were required to be 40 years of age or older, able to understand and sign informed consent, as well as have a best corrected visual acuity (BCVA) of 20/40 or better in both eyes. Exclusion from the study took place if subjects were unable to tolerate ophthalmic testing, had a history of complicated intraocular surgery, non-glaucomatous, visually impacting comorbidities, any neurodegenerative disease, any visually impacting disease apart from glaucoma or demonstrated unreliable VF/OCT testing, including poor fixation. Both eyes were evaluated, and a study eye was selected through randomization. The majority of subjects had both eyes simultaneously tested unless monocular VF testing was required due to reliability issues or comorbidities.

### 2.2. Study Design

The following tests were performed as part of the comprehensive exam: BCVA, slit lamp biomicroscope, IOP with a Goldmann tonometer, central corneal thickness (CCT), color fundus photography, 3D Wide (12 mm × 9 mm) OCT scan utilizing the Maestro2 (Topcon Corporation, Tokyo, Japan), and a threshold TEMPO AIZE-Rapid test (24-2, Stimulus Size III, Tracking OFF). OCT was used to identify glaucomatous structural defects and was considered reliable if the scan contained minimal artifacts and the image quality (TopQ) score was 25 or above. Structural defects were based on retinal nerve fiber layer thinning or ganglion cell loss as well as optic nerve head changes such as increased cup-to-disc ratio and neuroretinal rim thinning. Threshold VF testing with the TEMPO detected functional loss and was deemed reliable if fixation losses (FLs) were 20% or below, false positives (FPs) 10% or below, and false negatives (FNs) 12% and below. Functional loss was determined based on glaucoma-consistent defect patterns, with mean deviation and pattern standard deviation used to stratify disease loss severity. The threshold test results served as the ‘ground truth’ for classifying visual field outcomes as positive or negative.

Upon confirming all eligibility criteria, one TEMPO screening test (Stimulus Size III, Tracking OFF, Age corrected P1% stimulus threshold) was conducted by an ophthalmic technician in a dimly lit room and refractive corrective lenses were used if needed. Some patients only had the study eye tested with the threshold program due to the fellow eye not meeting inclusion or exclusion criteria. A second TEMPO screening test was then conducted to measure intra-session testing stability, with no defined break period between the two tests. A positive screening test was defined as missing at least one stimulus presentation at a test location, while a negative screening test was defined as detecting all presented stimuli.

### 2.3. Statistical Analysis

Sensitivity was assessed twice, once with the first test and once with the second test. To assess sensitivity, positive screening tests were divided by the total number of glaucoma patients included. Specificity was calculated by dividing the negative screening tests by the total number of healthy patients. Sensitivity was further analyzed for glaucoma subgroups based on mean deviation (MD) severity ranges: ≥−3 dB, −6 dB < −3 dB, and ≤−6 dB. MD was derived from the TEMPO threshold VF.

To evaluate the ability of the TEMPO screening test to detect the correct screening outcome, a logistic regression model was developed that accounts for ground truth, age of the patient, and the MD value. The outcome of the model is the screening test finding—a binary value of a positive or negative screening test. The data were randomly split into a 60% training and 40% testing data scenario for each repetition, applied to the logistic regression model, and repeated 50 times to increase precision of the estimates. Applying the exponential function to the coefficient estimates provided the odds of detecting true positive field defects. Descriptive statistics and logistic regression models were conducted using R (4.4.2). To improve the clarity and grammar of the manuscript, AI-based language enhancement tools, including ChatGPT (GPT-4-turbo, OpenAI, October 2024), were used in the manuscript revision process. However, all intellectual contributions and interpretations remain in the responsibility of the authors.

## 3. Results

### 3.1. Demographics and Clinical Characteristics

A total of 193 eyes (193 subjects) were evaluated in the final analysis (101 healthy and 92 glaucoma). Seven subjects were excluded due to visually impacting comorbidities and unreliable OCT and VF results. The mean age of the cohort was 66.6 years old [SD] [±12.7] (healthy mean age, 59.6 [±12.2]; glaucoma mean age, 74.3 [±8.1]). Mean screening test time per eye was 39.4 ± 4.9 s overall (healthy = 38.9 ± 4.5 s, glaucoma = 41.5 ± 6.2 s). The mean test time for the early, moderate, and advanced glaucoma subgroups were 40.0 ± 4.1 s, 45.3 ± 4.8 s, and 52.2 ± 11.3 s, respectively. Demographic and clinical characteristics are reported in Table 1.

### 3.2. Sensitivity and Specificity Measures

The first and second screening tests were analyzed as separate data cohorts and subgroups of MD categories further differentiated sensitivity measures. The specificity of the screening test was 91% (n = 101 eyes) for both tests (accuracy, 95% CI) (72%, 0.65, 0.78). The overall proportion of those with a positive field defect was 49% for both tests. For glaucoma cases with MD < −3 dB, sensitivity reached 100% in both tests. These findings are further summarized in Table 2.

### 3.3. Logistic Regression Model

MD (*p* < 0.05) and having glaucoma (*p* < 0.05) had significant association in detecting positive versus negative screening tests. Age did not have a significant association; therefore, it held no direct effect on outcome detection. The odds ratio of detecting a positive screening test was 4.637, indicating that the screening test is over four times more likely to correctly conclude a subject has a positive field defect than to incorrectly conclude that the eye is healthy after controlling for MD and age. For MD, the odds ratio was 0.497 indicating that for every 1 dB decrease in MD, an eye is 50% more likely to have a positive screening test. For the second test, the odds ratio for having glaucoma and MD was 6.09 and 0.472, respectively. Table 3 and Table 4 provide summarized model outputs.

## 4. Discussion

This study demonstrates that the TEMPO/IMOvifa screening program has high specificity, indicating a strong likelihood of accurate screening in an ocular-based screening environment with a predominantly healthy demographic. The specificity of the repeated suprathreshold tests was comparable, suggesting intra-session testing is reliable. These findings align with the specificity reported for previously validated suprathreshold strategies [15,16].

Sensitivity and its clinical relevance vary among MD subgroups. In a cohort with average MDs of ≥−3 dB, the sensitivity was lower compared to MDs < −3 dB, suggesting that the ability to detect a positive field defect with the TEMPO suprathreshold test is dependent on the severity of the disease. In testing applied to individuals whose MD was worse than −3 dB, the sensitivity measures showed excellent ability to screen for positive field defects. Confidence intervals show that the margin of error was slim. Screening tests are not usually considered to be diagnostic and are predominantly useful in identifying subsets of populations that require additional testing. Threshold testing may not always be a viable option in certain healthcare settings and utilizing a screening test as an initial baseline can expand patient testing volume, especially for those which may need it the most. Having a highly specific test is needed for ocular screening since majority of patients are likely without visual function loss. The bilateral nature of the TEMPO may contribute to specificity since there is improved comfort for the patient due to a patch not being needed which may minimalize test errors and false positives. The rapid test duration may reduce fatigue-related errors and improve reliability.

When interpreting our regression model, we find that it verifies which factors affect the screening test as well as the degree of the effects. The MD odd ratios indicate that lower values of MD were more likely to be in the glaucoma category, which is consistent with clinical expectations, and it also supports the expectation that MD has in being a predictor of a positive screening test. The odds ratio for the glaucoma category indicated that the model was four times more likely to correctly identify positive screening tests than negative screening tests after the first test, and six times more likely in the second test, demonstrating stability as testing is repeated. From this, the model implies that the screening test is more effective in detecting true positives than false positive cases (Table 3 and Table 4).

Previous works by Nishijima et al. [15] have focused on the ‘imo’ screening program (ISP), the predecessor of the TEMPO, by comparing its effectiveness to the frequency doubling technology (FDT). In line with our results, the ISP demonstrated high AUC and sensitivity in mild glaucoma populations and stronger AUC and sensitivity in moderate to severe glaucoma cases. Specificities of the TEMPO screening program were also higher than the values in FDT and ISP. They concluded that the screening program was thus comparable to the FDT and an effective tool in glaucoma screening. Our findings were also analogous to results in Arai et al., a work that examined the standalone diagnostic accuracy of the ‘imo’ screening device. Their AUC for mild, moderate, and severe glaucoma was 0.77, 0.97 and 1.0, respectively [14]. Sensitivity values for mild to severe glaucoma cases were also comparable to our results [14].

Other novel screening strategies have also been developed and tested in recent years. The glaucoma screening test developed for the Octopus perimeter (Haag-Streit, Koeniz, Switzerland) utilizes a 28-point suprathreshold program which tests points up to three times at a stimulus intensity less than 5%, designed for testing a large and diverse population [16]. They reported a sensitivity range of 93.9–100% in the moderate–severe glaucoma group, mirroring our reported findings for this range of glaucoma [16]. The timing for their healthy group was 40 s per eye and for early, moderate, and advanced glaucoma it was 55, 77, and 95 s per eye, respectively. The TEMPO screening test demonstrated faster timing than the Octopus, especially for the glaucoma group, with an average of 42 s (early = 40 s; moderate = 45 s; advanced = 52 s) per eye. For the healthy group, the TEMPO had an average time of 39 s. This is likely due to the fact that stimulus points are presented a maximum of two times in the TEMPO screening program. Similar to the screening pattern of the Octopus, the Humphrey Field Analyzer (HFA) employs a three-zone as well as a two-zone testing pattern. The HFA and TEMPO share a two-zone method but differ greatly due to the fact that the TEMPO tests both eyes simultaneously while HFA can only test one eye at a time. This dramatically affects test time and reduces burden on the patient.

Multi-sampling screening (MSS) techniques have also been used as a suprathreshold alternative to traditional SAP due to its comparable sensitivity with full-threshold strategies [17]. As seen in Artes et al., MSS focuses on bettering detection of defects in milder populations of glaucoma (−6 dB <) while also maintaining comparability to threshold strategies as MD worsens. Our findings are consistent with the screening accuracy of the MSS when accounting for mild to severe cases. It is important to note, however, that accuracy of glaucoma detection increases as the number of presentations increases [18,19]. This leads to improved diagnostic capability in milder cases when utilizing MSS. It can, however, be more burdensome to patients than other suprathreshold strategies due to more stimuli being repeated. Fatigue from the test could then lead to test-taking inaccuracies, especially when evaluating one eye at a time, as opposed to simultaneous bilateral evaluation.

Our tested screening program demonstrates a strong specificity (90%) in a healthy cohort and strong sensitivity in a moderate–severe glaucoma cohort, indicating its potential benefits as an ocular screener. However, a 90% specificity may still produce a considerable number of false positives in low-prevalence contexts. This highlights the need for a multi-tiered approach where positive screening cases are referred for further structural and functional testing. Enabling this care model may serve to standardize this screening tool across ocular care settings which in turn will lead to increased detection of disease especially when considered in high patient volume clinics. Public health initiatives may be able to leverage this technology to better approach high-risk populations such as older adults and individuals with a known family history of glaucoma.

A limitation our study faced is that majority of subjects had an MD ≥ −3 dB, which in turn could lower overall sensitivity. It should be considered, however, that general screening environments in the US predominantly include early cases of glaucoma while underserved regions may have a higher proportion of moderate to advanced—our study shows that we can sufficiently address both cases. We do aim to address this limitation by having proportionate and equal-sized cohorts of MD values in future work. A single-site study design may also limit the generalizability of our results, and future works will consider a multi-site approach. Additionally, the screening pattern discussed is only focused on the frequency of glaucomatous VF defects. Including other visually impacting diseases could prove beneficial. Age matching between the healthy and glaucoma cohorts was not performed because it could introduce age-related bias; however, structural and functional markers were used to characterize glaucoma in our study, which minimized the impact of age. In addition, stimulus intensity was adjusted for age, and hence age was already factored in when producing the test output. This is further shown in the linear regression model where contribution to age is not significant. Future studies should aim to isolate the age difference, however. Lastly, testing was conducted at a single site which consisted of a predominant Caucasian cohort, limiting the generalizability of our results. Future work will focus on screening a wider demographic range as well as further investigating different screening patterns and stimuli sizes and integrating structural data modalities into VF screening. Gathering repeatability measures of the screening program will also be conducted to further validate our model.

To conclude, our findings demonstrate a VF screening program capable of effectively screening for loss of visual function in mild to severe cases of glaucoma (MD < −3 dB). The screening pattern also showed a strong specificity in being able to accurately classify true healthy cases. Implementing this screening technology in optometric/ophthalmic settings that are routinely screening for glaucoma could lead to earlier vision loss detection in under screened populations and demographics. While it should not serve as a stand-alone diagnostic program due to the lower sensitivities in early glaucoma, this study suggests that it may serve as a useful screening tool when used with other diagnostic tests. The ease of testing and speed allows it to be a favorable option for both the practitioner and the patient.

## Figures and Tables

**Figure 1 jcm-14-01526-f001:**
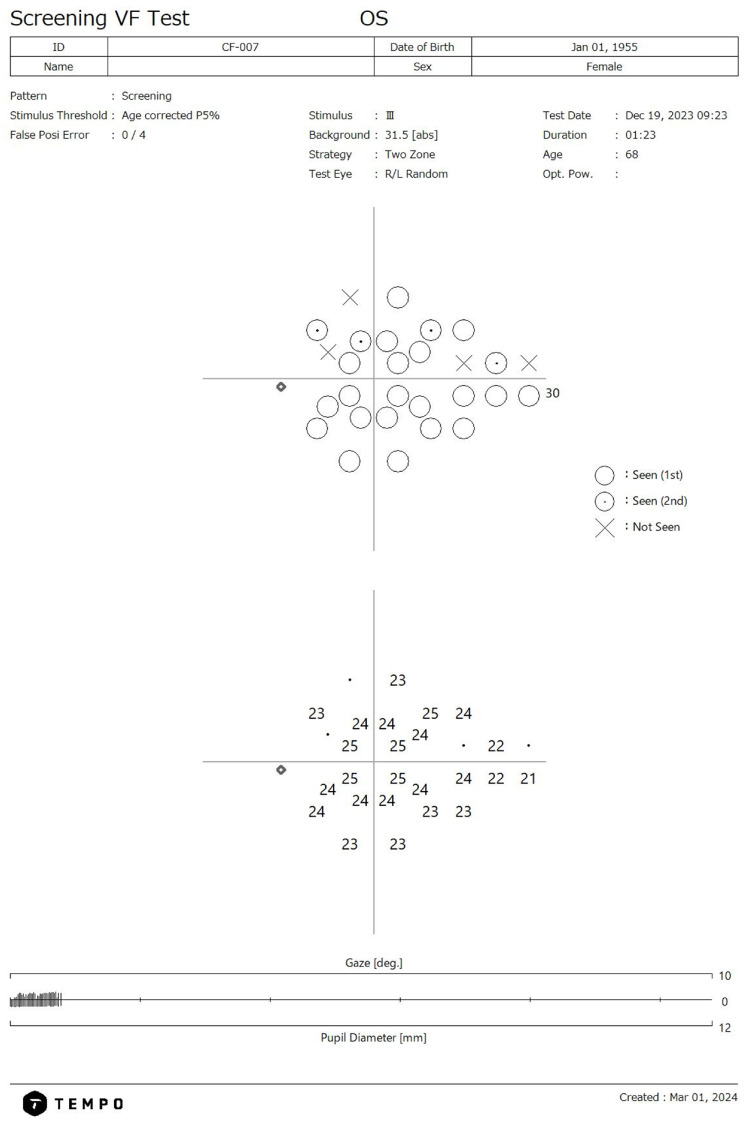
TEMPO screening report showing glaucomatous visual damage. A dotted circle indicates a stimulus was shown twice and an X indicates that the stimulus was not seen at that point. A non-dotted circle indicates the stimulus was seen on first presentation.

**Table 1 jcm-14-01526-t001:** Demographic and clinical characteristics.

	Overall (1)	Healthy (1)	Glaucoma (1)	*p*-Value (2)
Eyes (n)	193	101 (52.3%)	92 (47.7%)	
Mean Age (years)	66.6 ± 12.7	59.6 ± 12.2	74.3 ± 8.1	<0.001
Sex (Female)	118 (61.1%)	64 (63.3%)	54 (58.7%)	0.5
Race				0.7
White	142 (73.6%)	77 (76.2%)	65 (70.7%)	
Not disclosed	41 (21.2%)	20 (19.8%)	21 (22.8%)	
Black	9 (4.7%)	4 (4.0%)	5 (5.4%)	
Asian	1 (0.5%)	0 (0.0%)	1 (1.1%)	
Ethnicity				0.7
Hispanic	2 (1.0%)	1(1.0%)	1(1.1%)	
Non-Hispanic	146 (75.6%)	79 (78.2%)	67(72.8%)	
Not Disclosed	42 (21.8%)	20 (19.8%)	22 (23.9%)	
MD Category				<0.001
MD ≤ −6 dB (count)	8 (4.15%)	0 (0%)	8 (8.70%)	
−6dB < MD < −3 dB (count)	11 (5.70%)	3 (2.97%)	8 (8.70%)	
MD ≥ −3 dB (count)	174 (90.2%)	98 (97.0%)	76 (82.6%)	
Visual Field Indices				
Mean MD (dB)	−0.77 ± 3.0	0.08 ± 1.1	−1.7 ± 4.0	<0.001
Mean PSD (dB)	3.46 ± 13.14	1.63 ± 1.30	3.53 ± 3.23	<0.001
Mean Screening Test Time (s)	39.4 ± 4.9	38.9 ± 4.5	41.5 ± 6.2	<0.001
Clinical Indices				
CCT (µm)	558 ± 42	561 ± 41	554 ± 42	0.2
IOP (mm Hg)	15 ± 3	15 ± 3	15 ± 3	0.9
cpRNFL (µm)	89 ± 19	97 ± 18	79 ± 16	<0.001

(1) n (%); Mean (SD). (2) Fisher’s exact test; Wilcoxon rank-sum test; Pearson’s Chi-squared test.

**Table 2 jcm-14-01526-t002:** Summary of sensitivity and specificity results.

Data	Sensitivity	Specificity	Accuracy	95% CI Accuracy
1st test—ALL	0.49	0.91	0.72	(0.65, 0.78)
1st test with Pathology MD ≥ −3 dB	0.38	0.91	0.70	(0.63, 0.77)
1st test with Pathology −6 dB < MD < −3 dB	1.0	0.91	0.93	(0.86, 0.97)
1st test with Pathology MD ≤ −6 dB	1.0	0.91	0.93	(0.86, 0.97)
2nd test—ALL	0.49	0.91	0.72	(0.65, 0.78)
2nd test with Pathology MD ≥ −3 dB	0.37	0.91	0.69	(0.62, 0.76)
2nd test with Pathology −6 dB < MD < −3 dB	1.0	0.91	0.92	(0.85, 0.96)
2nd test with Pathology MD ≤ −6 dB	1.0	0.91	0.92	(0.85, 0.96)

**Table 3 jcm-14-01526-t003:** First test logistic regression model.

Coefficients	Estimate	Standard Error	Z-Value	*p*-Value
Intercept	−4.277	1.471	−2.907	0.004
Pathology Category	1.534	0.510	3.006	0.003
MD	−0.700	0.167	−4.196	<0.0001
Age	0.027	0.022	1.228	0.219

**Table 4 jcm-14-01526-t004:** Second test logistic regression model.

Coefficients	Estimate	Standard Error	Z-Value	*p*-Value
Intercept	−2.471	1.372	−1.802	0.072
Pathology Category	1.806	0.525	3.438	0.001
MD	−0.751	0.175	−4.296	<0.0001
Age	−0.002	0.022	−0.070	0.944

## Data Availability

Dataset available on request from the authors.

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
