# Peer review of "Validation of a Novel Suprathreshold Strategy for Screening Visual Function"

_jcm, 2025, doi:10.3390/jcm14051526_

Round 1
Reviewer 1 Report
Comments and Suggestions for Authors
The study entitled "Validation of a Novel Suprathreshold Strategy for Screening Visual Function" evaluates the efficacy of the screening program “TEMPO/IMOvifa suprathreshold" in the management of glaucoma. The authors have established a comprehensive framework for evaluating the specificity and sensitivity of this innovative screening method, therefore enhancing the knowledge of its clinical applicability.
The authors have effectively articulated the need for glaucoma screening, presenting a compelling justification for the advancement of rapid and dependable diagnostic methodologies, including suprathreshold tactics. The paper summarizes the shortcomings of current threshold procedures, especially their inadequacy for population-based screenings owing to their time-consuming nature and the risk of patient tiredness. This is crucial in establishing the need for the planned TEMPO screening program.
The research design has complied with the proper ethical and procedural criteria, including informed consent and respect to the Declaration of Helsinki. The criteria for subject inclusion and exclusion have been explicitly delineated, and the demographic distribution has been well characterized. The authors state the limits of the study, which are related to the cohort comprised mostly of Caucasian participants, which can have an impact on the generalizability of the conclusions. The study needs to be considered in different cohorts.
The results have been nicely documented. Statistical techniques used are proper to assess specificity and sensitivity. Logistic regression models have yielded significant insights into the predictive significance of several factors, including mean deviation (MD) and the presence of illness. The results have shown elevated specificity, which is essential for population-based screenings, and considerable sensitivity in instances of mild to severe glaucoma. The reduced sensitivity for early glaucoma, however, needs to be further addressed in diagnostic approaches.
The authors have properly compared their findings within the current literature, comparing them with other suprathreshold techniques and corroborating their results with previous investigations. This has highlighted the prospective benefits of the TEMPO screening program, notably its rapid testing duration and bilateral assessment capacity, which augment patient comfort and test dependability. The results show the program's congruence with clinical requirements for early diagnosis in marginalized groups.
Some limits should be mentioned. The manuscript might benefit from a more thorough examination of the significance of its results for clinical practice and healthcare policy. The implications of a single-site study design and techniques need to be addressed in further research, which needs to be mentioned. The paper could have considered additional detailed data on intra-session reliability for subgroups with differing illness severities to bolster the assertions of test stability. The paper needs minor editing to improve the flow. The tables and figures convey data properly. The references are suitable and coherent with the topic of the paper.
Author Response
Summary: Thank you for taking the time to review this manuscript and for providing comments. Please find the responses to each of your comments below. The revisions have been highlighted in red within the resubmitted manuscript file.
Comment #1: The manuscript might benefit from a more thorough examination of the significance of its results for clinical practice and healthcare policy.
Response #1: Thank you for commenting on this point. We recognize the value in it and have added a section to the discussion further discussing possible clinical practice and healthcare policy implications of our technology while also considering its capability in terms of specificity. This is a novel validation so we do want to refrain from making strong assumptions about policy change. Hopefully future studies using this screening program will further validate our current suggestions and lead to advantageous policy changes. Please see lines 256-265.
Comment #2: The implications of a single-site study design and techniques need to be addressed in further research, which needs to be mentioned.
Response #2: This is a valid point. We have added a mention of this limitation in the discussion as well as our plan to implement a multi-site approach in future studies. Please see lines 271-273.
Comment #3: The paper could have considered additional detailed data on intra-session reliability for subgroups with differing illness severities to bolster the assertions of test stability.
Response #3: Thank you for this comment and we agree that additional statistics may have strengthened test stability. The reason we chose to not include additional analysis was because it was not an intended objective of this study since the interval between test 1 and test 2 were not firmly defined so fatigue levels could be variable. We also believe that the very similar sensitivity and specificity between test #1 and test #2 shows it to be a repeatable test in nature. Future work will garner data on repeatability measures and enforce methods to enable this. We have added this as a future action to the limitation section. Please see lines 285-286.
Comment #4: The paper needs minor editing to improve the flow.
Response #4: We have improved certain language across the manuscript to improve the general flow and understandability of the manuscript. The methods section has also been restructured to improve understandability. Please see lines 41-44, 57-59, 166-167,192 and 205-206 for some specific changes.
Reviewer 2 Report
Comments and Suggestions for Authors
As the authors themselves state, the results of the study are suitable for screening and not for early diagnosis of glaucoma. That is why my rating is so high.
• What is the main question addressed by the research?
Design of a suprathreshold visual field program for glaucoma screening
• Do you consider the topic original or relevant to the field? Does it address a specific gap in the field? Please also explain why this is/ is not the case.
The topic is appropriately chosen, please note that it is mainly suitable for screening.
• What does it add to the subject area compared with other published material? Rapid screening for glaucoma disease.
• What specific improvements should the authors consider regarding the methodology?
The methodology does not need to be modified.
• Are the conclusions consistent with the evidence and arguments presented and do they address the main question posed? Please also explain why this is/is not the case.
Everything's fine. The proposed methodology is for rapid glaucoma screening.
• Are the references appropriate? no comments
• Any additional comments on the tables and figures.
no comments
Author Response
Summary:Thank you for taking the time to review this manuscript and for the positive note.
Reviewer 3 Report
Comments and Suggestions for Authors
Potentially a valuable paper but it needs some work (listed in approximate order of importance). Throughout, writing needs to be improved through thorough copy-editing (word choice, grammar, idiom); there are simply too many instances to list. I have encountered many sentences that are grammatically correct but make no sense at all (example, 56-57: “Two-zone…”).
Screening / screening test: this term has a defined meaning in medicine, quite different from the meaning of "visual field screening" in the old literature. Clarify, or avoid.
This is a diagnostic test evaluation, and there are well-established criteria for how such studies should be carried out and reported (STARD 2015 guidelines, and also see QUDAS-2, papers by PPM Bossoyt). Please review these and address the gaps where possible.
The authors evaluate the test results based on a single criterion for abnormality (2 stimuli missed on one test location). The paper would be much stronger if a other, more conservative criteria, could be added (to achieve a higher specificity, i.e. > 0.9).
I do not understand the rationale for the logistic regression analysis, or for the bootstrap “to increase the precision”. Please describe the model – what are the dependent and independent variables, and what is the objective?
There is a 15 year difference in age between the “controls” and the “glaucoma” participants. Do you see a problem with this, or why not?
Specific comments
Title 1: How is this strategy “novel”? To me, it looks quite similar to the suprathreshold strategies implemented on the original Humphrey Field Analyzer more than 40 years ago (if not slightly worse since only one intensity level is used). Previous tests (at least, HFA) should be described thoroughly and the differences discussed).
Ibid: Is “…screening visual function” a slight overstatement?
26-27: Please state clearly what you mean by “…conjunctive modalities to effectively diagnose” (or omit).
59-60: “…typically do not adjust…” As far as I’m aware, almost all ST tests increase the intensity once the initial presentations have been missed. Please clarify.
78: Is it a population, or rather a sample?
100: Please give clear criteria for structural and functional reference standards (see STARD)
120: how about “two stimulus presentations having been missed at the same test location…”. More generally, tidy up the termini to make the paper more easily understandable.
Table 1: There are separate rows for race and ethnicity – why not dispense with this nonsense altogether?
Table 1 ibid: Revise carefully the decimal places. Numbers in “MD category” surely should be integers, and also it does not make sense to quote CCT and RNFLt to the fraction of the wavelength of light.
152-153: “The overall proportion…” This is not how specificity is defined. Why are there 100 eyes, not 101?
175-176: “high specificity”. A specificity of 90% is nowhere near high enough to for population-based screening of glaucoma. There is a large literature on this so I will not rehearse the argument here. Do a simple thought experiment (1000 people to be screened, prevalence 2%, sensitivity = 75% and specificity =0.90: what will be the positive predictive value? ). This needs discussion.
260: “…shows it to be an effective screening tool.” Not by any stretch of the imagination. For a start, this would need a study in a proper screening setting (see Bossoyt etc). You could reword into “…suggests that it might be useful for screening when used with other diagnostic tests…” etc.
see above
Author Response
Summary: Thank you for taking the time to review this manuscript and for providing comments. Please find the responses to each of your comments below. The revisions have been highlighted in red within the resubmitted manuscript file.
Comment #1: Throughout, writing needs to be improved through thorough copy-editing (word choice, grammar, idiom); there are simply too many instances to list. I have encountered many sentences that are grammatically correct but make no sense at all (example, 56-57: “Two-zone…”).
Response 1: Thank you for pointing this out. We have gone through and made a number of modifications to word choice and sentence structure in order to improve understandability. The term ‘two-zone’ has been properly introduced and the methods section has been restructured to improve flow. The following changes have also been made across the manuscript: Please see lines 41-44, 57-59, 166-167, 192, 205-206.
Comment #2: Screening / screening test: this term has a defined meaning in medicine, quite different from the meaning of "visual field screening" in the old literature. Clarify, or avoid.
Response #2: We agree with your comment and have clarified the usage of the terms “screening” and “screening test” in the introduction to avoid confusion. Please see lines 50-52.
Comment #3: This is a diagnostic test evaluation, and there are well-established criteria for how such studies should be carried out and reported (STARD 2015 guidelines, and also see QUDAS-2, papers by PPM Bossoyt). Please review these and address the gaps where possible.
Response #3: Thank you for pointing this out. We have reviewed the mentioned guidelines and have made the following changes: Indicated the study design, location and nature of site where the study took place, made note that the threshold test was used as the ground truth, who operated the visual field testing as well as diagnostic criteria. We believe we have thoroughly met the other criteria listed by the cited guidelines. It should be noted that the purpose of this test is meant to identify potential cases for referral, which is a statement we have added to the introduction for enhanced clarity. Please see lines 78-80, 96-97, 117-119, 121-124, 126-127.
Comment #4: The authors evaluate the test results based on a single criterion for abnormality (2 stimuli missed on one test location). The paper would be much stronger if a other, more conservative criteria, could be added (to achieve a higher specificity, i.e. > 0.9).
Response 4: We agree that a more conservative criteria would strengthen results but it would not be representative of the current algorithm used by the commercially available device. Considering a positive test as two contiguous points not being seen rather than one point would indeed increase specificity but also decrease sensitivity.
Comment #5: There is a 15 year difference in age between the “controls” and the “glaucoma” participants. Do you see a problem with this, or why not?
Response #5: Thank you for pointing this out. The reason for this difference is due to glaucoma prevalence increasing with age. This difference may introduce age-related bias but it is counteracted by the fact that the diagnostic criteria for glaucoma was based on established structural and functional markers, minimizing the impact of age alone. In addition to this, the stimulus intensity is adjusted for age and hence age is already factored in when producing the test output, and the reason why older people are flagged more than younger people would need to be due to factors external to age alone. This is further shown in the linear regression model where the contribution to age is not significant which highlights one of the reasons, we decided to include this analysis. We have still addressed this in the limitations section. Please see lines 275-281.
Comment #6: Title 1: How is this strategy “novel”? To me, it looks quite similar to the suprathreshold strategies implemented on the original Humphrey Field Analyzer more than 40 years ago (if not slightly worse since only one intensity level is used). Previous tests (at least, HFA) should be described thoroughly and the differences discussed).
Response #6: We appreciate the reviewer’s perspective regarding the historical context of suprathreshold strategies. While it is true that suprathreshold perimetry has been used for decades, including the Humphrey Field Analyzer (HFA), our strategy differs. Our testing algorithm employs a two-zone method which also employs a simultaneous bilateral testing approach, which in itself is novel compared to other tests currently used clinically. This accelerates testing time greatly and also reduces the likelihood of fatigue and errors. We believe the usage of ‘novel’ should remain in the title. We have added information on the HFA suprathreshold pattern as well as why it is different than the TEMPO screening pattern in the discussion. Please see lines 240-244.
Comment #7: Ibid: Is “…screening visual function” a slight overstatement?
Response #7: We acknowledge the concern regarding the term; however, we believe this phrasing remains appropriate because our test evaluates a critical component of functional visual performance - namely, the ability to detect suprathreshold stimuli. While visual function encompasses multiple aspects such as visual acuity, contrast sensitivity, and depth perception, our test specifically screens for deficits in visual field sensitivity. Given that significant visual field loss directly impacts overall visual function, we find this terminology to be a justifiable representation of the test’s purpose.
Comment #8: 26-27: Please state clearly what you mean by “…conjunctive modalities to effectively diagnose” (or omit).
Response #8: Thank you for pointing this out. We agree this term can be confusing and have modified it. Please see line 26.
Comment #9 59-60: “…typically do not adjust…” As far as I’m aware, almost all ST tests increase the intensity once the initial presentations have been missed. Please clarify.
Response #9: We agree that this may have been overstated and have modified the sentence to indicate that some tests increase intensity while others may not. Please see lines 61-63.
Comment #10 78: Is it a population, or rather a sample?
Response #10: Thank you for pointing out this oversight. We have changed this to say sample rather than population. Please see lines 82-83.
Comment #11 100: Please give clear criteria for structural and functional reference standards (see STARD)
Response #11: Thank you for highlighting this gap. We have added clear criteria for structural and functional reference standards. Please see lines 117-119 and 121-123.
Comment #12: 120: how about “two stimulus presentations having been missed at the same test location…”. More generally, tidy up the termini to make the paper more easily understandable.
Response #12: Thank you for pointing this out. This specific description has been cleaned up and written in a cleaner manner. Please see lines 131-133. We have made additional changes to the termini of the paper. Please see Response #1 for details.
Comment #13: Table 1: There are separate rows for race and ethnicity – why not dispense with this nonsense altogether?
Response #13: Race and ethnicity are reported separately in accordance with clinical research standards (NIH, ICMJE) to ensure transparency in demographic differences. Given the established guidelines and the potential for disparities in glaucoma detection, we believe this approach is necessary and prudent.
Comment #14: Table 1 ibid: Revise carefully the decimal places. Numbers in “MD category” surely should be integers, and also it does not make sense to quote CCT and RNFLt to the fraction of the wavelength of light.
Response #14: We agree with your comment and have made changes to the number in “MD Category” and to CCT/RNFLt in Table 1.
Comment #15: 152-153: “The overall proportion…” This is not how specificity is defined. Why are there 100 eyes, not 101?
Response #15: Thank you for bringing these two points to our attention. We have reworded this sentence to properly define specificity and corrected 100 eyes to be the correct value of 101 eyes as seen in Table 1. Please see lines 164-165.
Comment #16: 175-176: “high specificity”. A specificity of 90% is nowhere near high enough to for population-based screening of glaucoma. There is a large literature on this so I will not rehearse the argument here. Do a simple thought experiment (1000 people to be screened, prevalence 2%, sensitivity = 75% and specificity =0.90: what will be the positive predictive value? ). This needs discussion.
Response #16: We understand your concern and thank you for commenting on it. We acknowledge that a 90% specificity may still have a significant number of false positives which is why we further elaborate on a multi-tiered approach within the manuscript. We have also removed mention of ‘population-based screening’ given your comment and have opted to orient this screener to address the ocular health space instead of primary care as well. Please see lines 256-265.
Comment #17: 260: “…shows it to be an effective screening tool.” Not by any stretch of the imagination. For a start, this would need a study in a proper screening setting (see Bossoyt etc). You could reword into “…suggests that it might be useful for screening when used with other diagnostic tests…” etc.
Response #17: We recognize that the previous sentence may have been overstating. We have adopted your recommendation and rephrased the sentence to better fit the study design. Please see lines 293-294.